# Increased Respiratory Drive after Prolonged Isoflurane Sedation: A Retrospective Cohort Study

**DOI:** 10.3390/jcm11185422

**Published:** 2022-09-15

**Authors:** Lukas Martin Müller-Wirtz, Dustin Grimm, Frederic Walter Albrecht, Tobias Fink, Thomas Volk, Andreas Meiser

**Affiliations:** 1Department of Anaesthesiology, Intensive Care and Pain Therapy, Saarland University Medical Center and Saarland University Faculty of Medicine, 66424 Homburg, Germany; 2Outcomes Research Consortium, Cleveland, OH 44195, USA

**Keywords:** intensive care, anesthesia, inhaled sedation, respiratory drive, isoflurane, propofol

## Abstract

Low-dose isoflurane stimulates spontaneous breathing. We, therefore, tested the hypothesis that isoflurane compared to propofol sedation for at least 48 h is associated with increased respiratory drive in intensive care patients after sedation stop. All patients in our intensive care unit receiving at least 48 h of isoflurane or propofol sedation in 2019 were included. The primary outcome was increased respiratory drive over 72 h after sedation stop, defined as an arterial carbon dioxide pressure below 35 mmHg and a base excess more than −2 mmol/L. Secondary outcomes were acid–base balance and ventilatory parameters. We analyzed 64 patients, 23 patients sedated with isoflurane and 41 patients sedated with propofol. Patients sedated with isoflurane were about three times as likely to show increased respiratory drive after sedation stop than those sedated with propofol: adjusted risk ratio [95% confidence interval]: 2.9 [1.3, 6.5], *p* = 0.010. After sedation stop, tidal volumes were significantly greater and arterial carbon dioxide partial pressures were significantly lower, while respiratory rates did not differ in isoflurane versus propofol-sedated patients. In conclusion, prolonged isoflurane use in intensive care patients is associated with increased respiratory drive after sedation stop. Beneficial effects of isoflurane sedation on respiratory drive may, thus, extend beyond the actual period of sedation.

## 1. Introduction

Sedation is a central treatment of intensive care, enabling life-saving invasive procedures such as mechanical ventilation. Inhaled isoflurane was recently approved for intensive care sedation in Europe based on a multicentric randomized trial [1]. Use of isoflurane is especially interesting for prolonged periods of sedation [2,3], as intravenous sedatives may accumulate or cause substantial harm after prolonged use [4,5,6].

Preclinical studies indicate that isoflurane increases respiratory drive [7,8,9]. Specifically, tidal volume and, thus, minute ventilation are better maintained with isoflurane than with propofol [9]. Consistently, patients sedated with isoflurane are more likely to breathe spontaneously than patients sedated with propofol despite moderate to deep sedation [10]. Own non-published clinical observations suggest that this effect may well extend beyond the actual period of sedation with patients showing increased respiratory drive after discontinuation of prolonged sedation with isoflurane. As an adequate respiratory drive after sedation stop is essential for a successful weaning of the patient from the ventilator, it is of considerable interest to investigate the post sedative effects of prolonged isoflurane use on ventilation.

This study, therefore, aims to investigate the post sedative effects of prolonged isoflurane use on ventilation in intensive care patients. Specifically, we hypothesized that isoflurane compared to propofol sedation for at least 48 h is associated with increased respiratory drive in intensive care patients over 72 h after sedation stop.

## 2. Materials and Methods

This study was approved by our Institutional Review Board with waived consent (approval date: 4 April 2022, reference number: 67/22, Ethikkommission der Ärztekammer des Saarlandes, Saarbrücken, Germany).

### 2.1. Study Design

This is a retrospective cohort study performed at a single academic center for surgical intensive care of the Saarland University Medical Center. We screened all patients ventilated for at least 96 h in 2019 for eligibility. All data were digitally extracted from the patient data management system (Copra, Version 5, Copra System, Berlin, Germany). Data were obtained from 48 h before sedation stop until 72 h after sedation stop.

### 2.2. Inclusion and Exclusion Criteria

Inclusion criteria were mechanical ventilation for at least 96 h with more than 48 h of continuous sedation with isoflurane or propofol as the primary sedative before sedation stop, at least three available blood gas analyses during spontaneous ventilation under sedation, at least 24 h of spontaneous ventilation and no re-sedation after sedation stop, and at least three available blood gas analyzes during spontaneous ventilation after sedation stop. Exclusion criteria were age < 18 years, switch of the sedative within 48 h before sedation stop, severe pulmonary diseases, death within the observation period, and patients under palliative care.

### 2.3. Drug Administration

Isoflurane (Isoflurane 100%, Piramal Critical Care, West Drayton, UK) was administered via the Sedaconda Anesthetic Conserving Device (ACD, Sedana Medical AB, Danderyd, Sweden) as recommended by the manufacturer. Briefly, the ACD was inserted between the endotracheal tube of the patient and the Y-piece of the breathing circuit of a common intensive care ventilator. The ACD was connected to a syringe pump (Perfusor compact, B. Braun, Melsungen, Germany) that delivered liquid isoflurane. A gas monitor (Vamos, Dräger Medical Deutschland GmbH, Lübeck, Germany) was connected to the ACD to monitor the end-tidal isoflurane concentration. Finally, a charcoal filter (FlurAbsorb, Sedana medical AB, Stockholm, Sweden) was connected to the expiratory port of the ventilator for gas scavenging.

Propofol 20 mg·mL^−1^ (Propofol Hexal, Hexal AG/Sandoz, Holzkirchen, Germany) was infused by a syringe pump (Perfusor Space, B. Braun, Melsungen, Germany) according to common clinical practice.

As natural for retrospective studies, there was no explicit protocol for sedation. However, a written standard operating procedure of our center (provided as Appendix A) stipulates to administer sedative drugs as low as possible according to the patient’s needs and to perform daily spontaneous awakening trials for avoidance of overdosing and assessment of neurological function.

### 2.4. Ventilation

Patients were ventilated with Evita 4 ventilators (Dräger Medical Deutschland GmbH) in pressure-controlled mode (biphasic positive airway pressure) or pressure-support mode. Ventilation parameters were automatically captured by our patient data management system. In the patients that were extubated after sedation stop, ventilation parameters were captured from periods of non-invasive ventilation via a face mask.

### 2.5. Measurements

All available blood gas analyzes (BGA) within the observation period while patients were breathing spontaneously were included to evaluate respiratory drive. Circulatory and ventilatory measures were extracted from the patient data management system at 12-h intervals. Implausible values, as commonly obtained during periods of nursing or airway leaks, were excluded. Intravenously and orally applied opioids were converted to morphine equivalent doses (µg/kg) as previously published [11] to enable comparison (sufentanil 1:1000; hydromorphone 1:7; remifentanil 1:200). For remifentanil, the equivalent dose was divided by 60 to account for the considerably shorter half-life than morphine. The sum of all morphine equivalent doses over 12-h intervals was divided by 12 to obtain morphine equivalent dose rates (µg/kg/h). Patients that received additional opioid boluses from nurse-controlled analgesia pumps not being electronically recorded after sedation stop were excluded from the analysis of opioid consumption. The Simplified Acute Physiology Score II (SAPS II) was calculated according to Le Gall et al. [12]. The Sequential Organ Failure Assessment (SOFA) score was calculated according to Vincent et al. [13]. Ideal body weight was calculated according to the sex-specific ARDSnet formulas [14].

### 2.6. Outcomes

The primary outcome was increased respiratory drive after sedation stop, defined as arterial carbon dioxide pressure < 35 mmHg and base excess >−2 mmol/L to exclude potential respiratory compensations of metabolic acidosis. Secondary outcomes were measures of acid-base balance and ventilation including pH, arterial carbon dioxide partial pressure, base excess, tidal volume, respiratory rate, and inspiratory pressure support.

### 2.7. Statistical Analysis

Data were collected with Excel Version 16.58 (Microsoft, Redmond, WA, USA). Statistical analyses were carried out with R (*v4.0.2*, R Core Team, 2020) using the packages *readxl* (*v1.3.1*, Wickham and Bryan, 2019), *dplyr* (*v1.0.5*, Wickham, François, Henry, and Müller, 2021), *tableone* (*v0.12.0*, Yoshida and Bartel, 2020), *rcompanion* (*v2.4.1*, Mangiafico, 2016), *geepack* (*v1.3-2*; Højsgaard, Halekoh, and Yan, 2006), *parameters* (*v0.14.0*; Lüdecke, Ben-Shachar, Patil and Makowski, 2020), and *ggplot2* (*v3.3.3*; Wickham, 2016).

Normality was assessed by visual assessment of histograms/quantile-quantile plots and Shapiro–Wilk testing. According to data distribution, we present continuous measures as means with standard deviations or medians with interquartile ranges (IQR) for descriptive data and with the corresponding 95% confidence intervals (95% CI) for outcome data. Categorical variables are presented as frequencies (percentages).

Baseline balance is presented as absolute standardized differences, defined as the absolute difference in means divided by the pooled standard deviation. Repeated-measures data were summarized with a mean for each patient for the periods of 48 h before and 72 h after sedation stop and compared between groups by independent samples t-tests or Wilcoxon rank-sum tests. A two-sided *p* < 0.05 was considered statistically significant.

The risk ratio for increased respiratory drive in isoflurane versus propofol-sedated patients was calculated by Poisson generalized estimating equation regression to account for repeated measures. Two separate univariable models were calculated to estimate the crude risk ratio before and after sedation stop. Multivariable models were calculated to adjust for age, total ventilation and sedation time, tracheostomy, hemodialysis, simplified acute physiology score II, and mean morphine equivalent dose rate.

To our knowledge, there are no previous data on the prevalence of increased respiratory drive after sedation stop in intensive care patients. We, therefore, did not estimate sample size in advance and planned to include all qualifying patient records from a one-year cohort.

## 3. Results

### 3.1. Study Population Characteristics

A total of 158 patients were ventilated for at least 96 h throughout 2019. After application of inclusion and exclusion criteria, 23 patients sedated with isoflurane and 41 with propofol were included (Figure 1).

Potential covariates/confounders for respiratory drive including age, total ventilation and sedation times, tracheostomy, hemodialysis, and simplified acute physiology score II were not well balanced between the sedation groups (Table 1), and the analysis of increased respiratory drive was, therefore, adjusted for these variables.

Circulatory measures were similar in both groups; only heart rate was significantly higher in isoflurane patients during sedation (mean [95% CI]: isoflurane: 95 [87, 101], propofol: 84 [79, 89], *p* = 0.012; Table 2).

Sedatives were applied within a low dosing range with isoflurane applied at around 0.5 minimum alveolar concentration (MAC) and propofol applied below 2 mg/kg/h (Table 2). Opioid consumption was similar with both sedatives before and after sedation stop (Table 2). During sedation, patients received continuous intravenous opioids, either remifentanil, sufentanil, or hydromorphone. After sedation stop, in most patients, the continuous opioid infusion was stopped, and oral opioids or intravenous opioid boluses were applied. Thirty percent (7/23) of patients after isoflurane and 39% (16/41) after propofol sedation received occasional boluses of intravenous opioids via a nurse-controlled analgesia system not being electronically recorded and were, therefore, excluded from the analysis of opioid consumption after sedation stop.

### 3.2. Primary Outcome—Increased Respiratory Drive after Sedation Stop

We detected increased respiratory drive at 31% (159/515) of the observations in isoflurane-sedated patients compared to only 12% (110/924) in propofol-sedated patients within 72 h after sedation stop. Patients sedated with isoflurane were three times as likely to show increased respiratory drive within 72 h after sedation stop than those sedated with propofol: risk ratio [95% CI]: 2.6 [1.3, 5.2], *p* = 0.005, which remained similar after adjustments for age, total ventilation and sedation times, tracheostomy, hemodialysis, and simplified acute physiology score II: adjusted risk ratio [95% CI]: 2.9 [1.3, 6.5], *p* = 0.010 (Table 2, Figure 2). Additional adjustment for the mean morphine equivalent dose rate for those patients with complete data on opioid intake did not substantially change the association: adjusted risk ratio [95% CI]: 3.3 [1.3, 8.3], *p* = 0.012 (isoflurane: *n* = 16, propofol *n* = 25).

In contrast, increased respiratory drive was equally frequent with both sedatives before sedation was discontinued: adjusted risk ratio [95% CI]: 0.9 [0.2, 5.4], *p* = 0.925 (Table 2, Figure 2).

### 3.3. Secondary Outcomes—Acid-Base Balance, Ventilation and Opioid Consumption

There was no difference in blood pH between isoflurane-sedated and propofol-sedated patients (Table 2). However, base excess was significantly higher in isoflurane-sedated patients before sedation stop (median [95% CI]: isoflurane: 4.3 [3.1, 5.0], propofol: 1.9 [0.7, 2.7], *p* = 0.005; Table 2), suggesting metabolic compensation of slightly increased arterial carbon dioxide partial pressures before sedation stop (median [95% CI]: isoflurane: 47 [44, 52], propofol: 44 [42, 45], *p* = 0.096; Table 2, Figure 3). Although tidal volumes and respiratory rate did not substantially change after sedation stop, arterial carbon dioxide partial pressure was significantly lower in patients sedated with isoflurane compared to those sedated with propofol after sedation stop (median [95% CI]: isoflurane: 37 [35, 42] mmHg, propofol: 41 [39, 45] mmHg, *p* = 0.007; Table 2, Figure 3).

Tidal volumes were about 100 mL greater in patients sedated with isoflurane than in those sedated with propofol, with nearly identical differences before and after sedation stop (mean [95% CI]: before sedation stop: isoflurane: 613 [559, 660], propofol: 526 [503, 550], *p* = 0.001; after sedation stop: isoflurane: 609 [556, 668], propofol: 503 [471, 540], *p* = 0.002; Table 2, Figure 2). Respiratory rate and inspiratory pressure support did not differ significantly between the sedation groups (Table 2, Figure 3).

## 4. Discussion

Isoflurane-sedated patients were about three times as likely to have an arterial carbon dioxide pressure below 35 mmHg during periods of spontaneous breathing after sedation stop than those sedated with propofol. A comparatively better-maintained respiratory drive with isoflurane as opposed to propofol sedation, thus, seems to extend to the post-sedation period. To our knowledge, this is the first report on differential post-sedative effects of prolonged inhaled versus intravenous sedation on ventilation in intensive care patients.

In line with our primary finding, studies in rats showed that minute ventilation is better maintained with isoflurane than with propofol [9]. Most interestingly, isoflurane even increases respiratory drive at subanesthetic doses of 0.5 MAC but decreases respiratory drive at doses exceeding 1 MAC in rats [9]. Consistently, subanesthetic doses of volatile anesthetics promote the transition from controlled to spontaneous ventilation in intensive care patients [10,15,16]. Our study, thus, adds to current evidence that better maintenance of respiratory drive in isoflurane-sedated patients may continue well beyond the actual period of sedation.

Brainstem neurons of the retrotrapezoid nucleus are responsible for the maintenance of spontaneous breathing under general anesthesia [17]. The increased respiratory drive under isoflurane compared to propofol sedation can be explained by diverging effects on the central regulation of breathing; whereby propofol inhibits, isoflurane stimulates neurons of the retrotrapezoid nucleus, and, thus, increases respiratory drive [7,8].

The largest amount of isoflurane is exhaled during the first hours after anesthesia [18,19]. However, modern highly sensitive analytical methods show that volatile anesthetics are exhaled up to two weeks after general anesthesia [20,21]. Isoflurane trace concentrations were even detected in breath up to 130 days after anesthesia [21]. Whereas general anesthesia may last a few hours, we included patients exposed to isoflurane over at least 48 h. Therefore, additional saturation of body tissues with isoflurane leading to even longer final elimination times can be assumed. Residual pharmacologically active isoflurane concentrations could, thus, explain the observed increased respiratory drive after sedation stop in patients exposed to prolonged isoflurane sedation.

Arterial carbon dioxide pressure was slightly higher in isoflurane-sedated patients within the last 48 h before sedation stop, as opposed to being significantly lower within 72 h after sedation stop when compared to propofol-sedated patients. Slightly higher arterial carbon dioxide pressures under isoflurane sedation are explainable by the fact that volatile anesthetic reflection devices increase respiratory dead space resulting in carbon dioxide retention [22,23,24,25]. Interestingly, arterial carbon dioxide pressures dropped considerably after sedation stop in patients sedated with isoflurane, while tidal volume and respiratory rate remained almost unchanged, which is consistent with reduced dead space ventilation after removal of the volatile anesthetic reflection device.

Compared to patients sedated with propofol, tidal volumes were higher in patients sedated with isoflurane before and after sedation. While a compensatory increase in tidal volume may be a consequence of increased ventilatory dead space under isoflurane sedation [22,23,24,25], tidal volumes remained higher after sedation stop, even though volatile anesthetic reflection devices were removed. Animal data suggest that an isoflurane-induced increase in minute ventilation is largely caused by increased tidal volumes [9]. It, thus, seems likely that the observed phenomenon results from residual isoflurane concentrations increasing respiratory drive. However, this could also reflect a physiological consequence of prolonged exposure of the lung to higher tidal volumes causing adaptations in the neural control of breathing. Consistent with this theory, a lung volume-related habituation and desensitization of the Hering–Breuer inflation reflex was shown in rats [26]. In general, both causes are interesting with potential clinical consequences and should be subject to future studies.

Our study has distinct limitations. At first, age, total ventilation and sedation time, tracheostomy, hemodialysis, and simplified acute physiology score II differed markedly between the sedation groups. However, all of them only marginally influenced the primary outcome—increased respiratory drive after sedation stop. Furthermore, uncaptured adjunct drugs, such as benzodiazepines, may have influenced respiratory drive and ventilatory measures, although unlikely since our center largely dispenses with the administration of benzodiazepines. Of note, our study represents an initial observation of increased respiratory drive after discontinuation of prolonged isoflurane sedation. Future studies with larger sample sizes and distinct treatment protocols for sedation, opioid use, and adjunct drugs may provide more accurate estimates of differences in respiration between isoflurane and propofol sedation.

## 5. Conclusions

Prolonged isoflurane use in intensive care patients was associated with increased respiratory drive throughout 72 h after sedation stop. Beneficial effects of isoflurane sedation on respiratory drive may, thus, extend beyond the actual period of sedation. However, these results still need confirmation by studies with larger sample sizes and at best by prospective investigations.

## Figures and Tables

**Figure 1 jcm-11-05422-f001:**
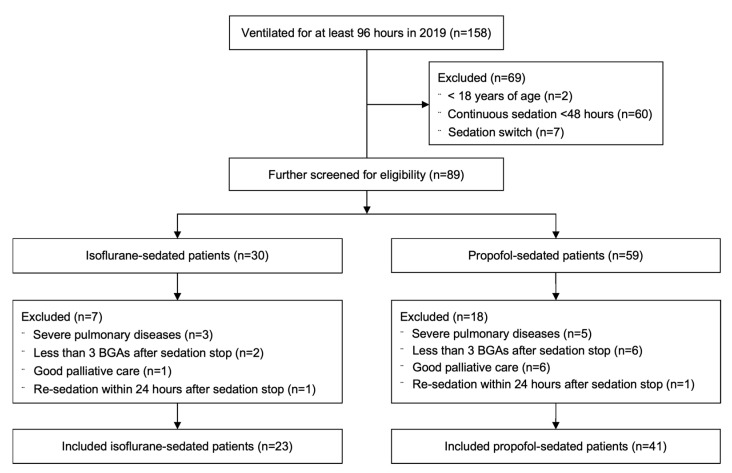
Patient flow chart.

**Figure 2 jcm-11-05422-f002:**
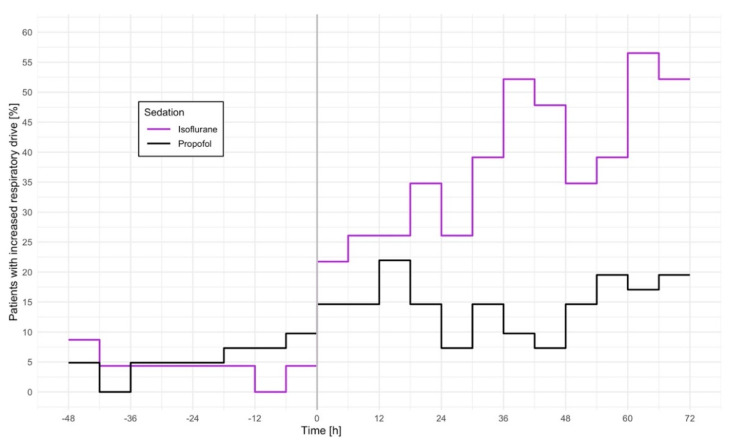
Percentage of patients with increased respiratory drive. Sedation was stopped at time point 0. Data are presented in 6-hour intervals as percentage of patients within each sedation group. The risk for increased respiratory drive after sedation stop was three times higher in patients sedated with isoflurane than in those receiving propofol: adjusted risk ratio [95%CI]: 2.9 [1.3, 6.5], *p* = 0.010. Increased respiratory drive was defined as arterial carbon dioxide partial pressure (P_a_CO_2_) < 35 mmHg and base excess > −2 mmol/L.

**Figure 3 jcm-11-05422-f003:**
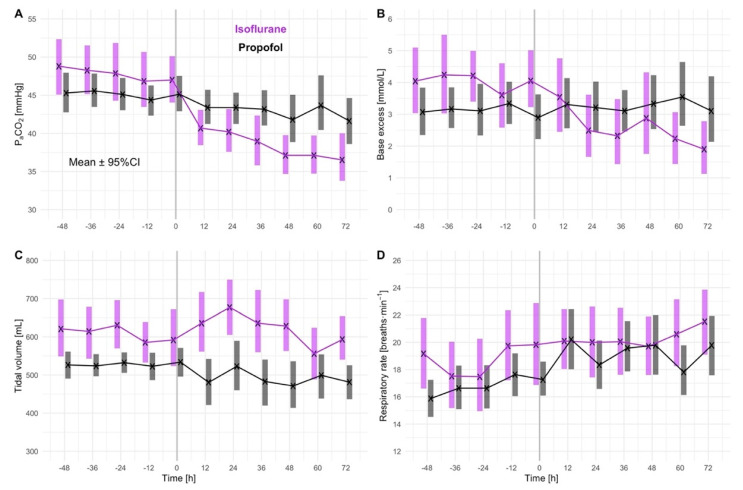
Acid-base balance and ventilation. Sedation was stopped at time point 0. Data are presented in 12-hour intervals as means ± 95% confidence intervals (95%CI). PaCO_2_, arterial carbon dioxide pressure. (**A**) Arterial carbon dioxide partial pressure; (**B**) Base excess; (**C**) Tidal volume; (**D**) Respiratory rate.

**Table 1 jcm-11-05422-t001:** Study population characteristics.

Parameter	Isoflurane	Propofol	SMD
*n*	23	41	-
Sex [male]	20 (87)	24 (58)	0.673
Age [years]	55 [52, 65]	69 [60, 80]	0.833
Height [cm]	175 [171, 180]	170 [165, 178]	0.218
Weight [kg]	85 ± 28	81 ± 23	0.158
BMI	26 [23, 32]	27 [23, 30]	0.028
SAPS II	37 ± 13	41 ± 13	0.313
SOFA	10 ± 4	10 ± 3	0.121
CVVHD [*n*]	11 (48)	11 (27)	0.445
Death [*n*]	5 (22)	13 (32)	0.227
Tracheostomy [*n*]	16 (70)	11 (27)	0.946
Total ventilation time [h]	114 [86, 171]	108 [79, 167]	0.402
Total sedation time [h]	179 [141, 234]	108 [79, 167]	0.845
Surgical patients [*n*]	20 (87)	38 (93)	
Visceral [*n*]	11 (48)	19 (46)	-
Trauma [*n*]	2 (9)	10 (24)	-
Other [*n*]	7 (30)	9 (22)	-
Medical patients [*n*]	3 (13)	3 (7)	-

Data are reported as means ± standard deviations, medians [interquartile ranges], or numbers (percentages). The standardized mean difference (SMD) is presented as a measure of balance. BMI, body mass index. SAPS II, Simplified Acute Physiology Score II (scored at intensive care unit admission). SOFA, sepsis-related organ failure assessment score (scored 24 h before sedation stop). CVVHD, continuous veno-venous hemodialysis.

**Table 2 jcm-11-05422-t002:** Circulatory and ventilatory measures within 48 h before and 72 h after sedation stop.

	Before Sedation Stop			After Sedation Stop		
Parameter	Isoflurane	Propofol	P	Isoflurane	Propofol	P
*n*	23	41	-	23	41	-
**Circulation**						
Heart rate [bpm] *	95 [87, 101]	84 [79, 89]	**0.012**	93 [86, 100]	87 [83, 92]	0.157
Mean arterial blood pressure [mmHg]	69 [61, 69]	71 [68, 73]	0.195	80 [71, 88]	78 [72, 80]	0.585
**Sedation and analgesia**						
End-tidal isoflurane [Vol%]	0.64 [0.55, 0.70]	-	-	-	-	-
Propofol dose [mg/kg/h]	-	1.4 [1.1, 1.7]	-	-	-	-
Morphine equivalent dose [µg/kg/h]	39 [29, 60]	31 [22, 38]	0.087	34 [18, 46] ^†^	25 [15, 26] ^†^	0.073
**Primary outcome—Increased respiratory drive**						
Total observations [*n*]	318	520		515	924	
Observations with increased respiratory drive [*n*]	9 (3%)	27 (5%)		159 (31%)	110 (12%)	
Risk ratio	0.5 [0.1, 2.1]		0.319	2.6 [1.3, 5.2]		**0.005**
Adjusted risk ratio ^#^	0.9 [0.2, 5.5]		0.926	2.9 [1.3, 6.5]		**0.010**
Adjusted risk ratio ^#,§^	0.9 [0.2, 5.4]		0.925	3.3 [1.3, 8.3] ^†^		**0.012**
**Secondary outcomes—Acid–base balance and ventilation**						
pH	7.41 [7.38, 7.43]	7.40 [7.37, 7.41]	0.374	7.45 [7.41, 7.46]	7.43 [7.42, 7.44]	0.221
P_a_CO_2_ [mmHg]	47 [44, 52]	44 [42, 45]	0.096	37 [35, 42]	41 [39, 45]	**0.007**
Base excess [mmol/L]	4.3 [3.1, 5.0]	1.9 [0.7, 2.7]	**0.005**	1.7 [0.9, 3.0]	2.5 [1.5, 3.5]	0.297
Tidal volume [ml] *	613 [559, 660]	526 [503, 550]	**0.001**	609 [556, 668]	503 [471, 540]	**0.002**
Tidal volume normalized to IBW [ml/kg] *	9.0 [8.4, 9.6]	8.2 [7.8, 8.5]	**0.014**	9.0 [8.3, 9.7]	7.8 [7.3, 8.2]	**0.006**
Respiratory rate [bpm]	17 [15, 19]	17 [15, 17]	0.238	19 [15, 19]	19 [16, 20]	0.445
Inspiratory pressure support [cmH_2_O] *	8 [7, 10]	8 [7, 9]	0.377	6 [5, 7]	7 [6, 8]	0.362

Repeated measures were summarized with a mean for each patient and are reported as means (*) or medians with the corresponding 95% confidence intervals (95% CI) for each sedation group within 48 h before and 72 h after sedation. Groups were compared using independent samples t-tests or Wilcoxon rank-sum tests. Statistical significances (*p* < 0.05) are written in bold. The presented risk ratios [95% CI] were calculated by Poisson generalized estimating equation regression and describe the effect of isoflurane versus propofol sedation on increased respiratory drive within 48 h before or within 72 h after sedation stop. Increased respiratory drive was defined as arterial carbon dioxide partial pressure < 35 mmHg and base excess > −2 mmol/L. ^#^ Adjusted for age, total ventilation and sedation time, tracheostomy, hemodialysis, and simplified acute physiology score II. ^§^ Additional adjustment for mean morphine equivalent dose rate. ^†^ 30% (7/23) of patients after isoflurane and 39% (16/41) after propofol sedation were excluded due to opioid intake via a nurse-controlled analgesia system, which was not electronically recorded. P_a_CO_2_, arterial carbon dioxide partial pressure. SpO_2_, oxygen saturation by pulse oximetry. IBW, ideal body weight.

## Data Availability

The datasets analyzed during the current study are available from the corresponding author on reasonable request.

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
