# Peer review of "Increased Respiratory Drive after Prolonged Isoflurane Sedation: A Retrospective Cohort Study"

_jcm, 2022, doi:10.3390/jcm11185422_

Round 1
Reviewer 1 Report
The authors performed a retrospective cohort study to investigate if isoflurane sedation was associated with an increase in respiratory drive after sedation in mechanically ventilated adults in the ICU. The study may be useful to show a potential advantage of using isoflurane in the ICU. However, current manuscript as it is may not clinically useful since the outcomes are limited to the respiratory parameters, and the length of the ICU stay, the length of mechanical ventilation, etc. are not included. If the authors are purely interested to show the difference of the two anesthetics (isoflurane and propofol) in the ventilatory response, the manuscript still fails to achieve the goal due to insufficient information discussed in the following.
-Although this is not a randomized study nor a prospective study, to compare the ventilatory effects of the two sedatives (anesthetics), the sedation level of the patients and how the sedatives were titrated to the effect must be included. Without this key information, the effect on the ventilatory responses are not useful. In addition, isoflurane concentrations (end-tidal concentrations) and the propofol infusion rates are essential, both of which appear to be missing.
-What is the definition of “respiratory drive” in this manuscript? The respiratory drive indicates the intensity of the output from the respiratory centers. Clinically that reflects the respiratory (ventilatory) response to the levels of carbon dioxide (through chemoreceptors). This study did not measure this ventilatory response.
-The authors states, “better maintenance of respiratory drive in the isoflurane-sedated patients” and “residual pharmacologically active isoflurane concentrations could thus explain the observed increased respiratory drive after sedation stop…” and conclude that prolonged isoflurane use in intensive care patients was associated with increased respiratory drive. The manuscript simply shows that PaCO2 was significantly lower and the tidal volume was significantly larger in the patients after isoflurane sedation compared to propofol. In other words, it can be concluded that ventilatory suppression was prolonged after propofol sedation while the tidal volume and PaCO2 were recovered faster after isoflurane sedation. Whether the data show “increased” ventilatory response after isoflurane or “quick recovery” of ventilatory response after isoflurane is not distinguishable based on the data provided.
-The data handling and presentation need clarification. Table 2 shows “before” and “after” data. Are these averaged (at least) 3 blood gas data? When were these data obtained? Opioids are shown as morphine-equivalent, which is helpful. However, the opioids used include very short-acting and long-acting, which could influence the ventilatory parameter differently. Are there other adjunct drugs, such as benzodiazepine, alpha2-adrenergic agonist, etc.?
-How the hypothesis that “isoflurane compared to propofol sedation for at least 48 hours is associated with increased respiratory drive in the intensive care patients over 72 hours after sedation stop” is generated is not clear, especially why “72 hours” was selected. If this is based on the authors’ “own observation”, the observation needs to be discussed. During the measurements “after” sedation stop, were patients still intubated? When were they extubated?
Reviewer 2 Report
This manuscript entitled “Increased respiratory drive after prolonged isoflurane sedation: a retrospective cohort study” tested the hypothesis that isoflurane compared to propofol sedation for at least 48 hours is associated with increased respiratory drive in intensive care patients over 72 hours after sedation stop. This retrospective study was conducted at a single academic center for surgical intensive care. The authors demonstrated that stimulating effects of isoflurane on respiratory drive may extend beyond the actual period of sedation. This manuscript was well written. However, I have several concerns:
1. The author mentioned that in those patients that were extubated after sedation stop, ventilation parameters were captured from periods of non-invasive ventilation via a face mask. How did the author confirm the accuracy of the parameters of the non-invasive ventilator, especially in unintentional leaks situation?
2. The author mentioned that the patients receiving at least 48 hours of isoflurane or propofol sedation were compared. How does the author judge the level of sedation? Is the level of sedation consistent between the two groups of patients?
3. The author mentioned an increase in respiratory drive after sedation stop in patients with isoflurane sedation. Could the increase in respiratory drive be the result of the patient returning to a physiological state? Has the author compared respiratory drive after complete elimination of sedative effects?
4. Isoflurane will cause dose-related respiratory depression and decreased carbon dioxide ventilation. In the author's Figure 2, arterial carbon dioxide pressure increases, why does the base excess increase?
Reviewer 3 Report
Thank you for the opportunity to review this article.
I think that this is a high-quality study, however, there are some aspects need to be revised.
First of all, there is a lack of data about the diseases of enrolled patients. The respiratory drive could be affected by the characteristics of each disease.
Second, I think that opioid dose should be included in the multivariable analysis model. The authors documented that isoflurane use is associated with increased respiratory drive after sedation stop. However, the opioid dose is not included in multivariable model. The independent effect of isoflurane on respiratory drive regardless of opioid dose may enhance the clarity of the study.
[Minor]
Page 7, Line 191~194 should be removed.
Round 2
Reviewer 2 Report
I have no more comments.
Author Response
Thank you for your outstanding help in improving our manuscript.